# Study of the Sediment Transport Law in a Reverse-Slope Section of a Pressurized Pipeline

**Jiayi Wang [1,2], Yitian Li [1], Li Pan [2], Zhiqiang Lai [2,3] and Shengqi Jian [3,4,*]**

[1] School of Water Resources and Hydropower Engineering, Wuhan University, Wuhan 430000, China; hkywangjiayi@163.com (J.W.); wuhantianyi412@sina.com (Y.L.)

[2] Yellow River Institute of Hydraulic Research, Yellow River Conservancy Commission, Zhengzhou 450003, China; xiaogang4081@126.com (L.P.); zhshirley89@sina.com (Z.L.)

[3] Key Laboratory of Yellow River Sediment Research, Ministry of Water Resources, Zhengzhou 450003, China

[4] College of Water Conservancy Science and Engineering, Zhengzhou University, Zhengzhou 450003, China

* Correspondence: jiansq@zzu.edu.cn

**Abstract:** This article reveals the change law of the head loss and critical deposition velocity during hydraulic transmission of a solid–liquid two-phase pipeline. This article also establishes a physical test model. A single variable is used to conduct the experimental research by changing the conditions of the pipeline flow rate, the sediment concentration, and the reverse slope degree. Based on an analysis of the test process, a new formula is proposed to determine the critical sedimentation rate of the pipeline that considers a change in the adverse slope. By analyzing the variation rule of the hydraulic slope of the pipeline sediment in different states and comparing the hydraulic slope of the horizontal pipeline and reverse pipeline in different states, different factors that influence head loss are revealed. Finally, the measured value of this test is compared with the Durand equation and the Worster equation. It was found that the measured value of this test was more similar to the Durand equation. This study not only provides theoretical support for sand removal in pipelines but also promotes sedimentation in reservoirs.

**Keywords:** inverse slope pipeline; critical flow velocity; reservoir sedimentation; hydraulic gradient

## 1. Introduction

Reservoir projects on sandy rivers inevitably face the problem of sediment deposition [1,2]. Sedimentation influences the normal operation and service life of a reservoir, which causes serious events, such as the abandonment of reservoirs and dam breaking accidents [3–6]. In the Yellow River Basin and northwest China, in particular, reservoir siltation has been the most prominent problem [7–9]. Reservoir sedimentation and desilting have become urgent problems and have attracted the attention of governments and water conservancy departments.

To reduce siltation in reservoirs, many reservoirs discharge silted sediment out of the reservoir by means of flood discharge, a density current, and the emptying of reservoirs. In addition, the water head difference between the upper and lower reaches of the reservoir is used to discharge sediment in pipelines. [10]. Sediment delivery is restricted by the topography of the sediment in front of the dam, inflow flow, outflow flow, fine sediment content, and the water level in front of the dam. The discharged sediment can be only a portion of the retained sediment in front of the dam. Pipeline sediment delivery is more flexible than reservoir operations, the desilting area can be expanded, and the desilting effect can be improved by reasonably arranging the desilting pipeline [11–14].

Self-priming pipeline sediment discharge is an engineering technique for reservoir pipeline sediment discharge that uses the natural water head of the reservoir and does not need external

dynamic conditions [15,16]. An underwater pipe drainage system with a suction head is installed in the reservoir area to discharge the sediment deposited in the reservoir area. A reservoir self-priming pipeline uses the natural water head of the reservoir without external dynamic conditions, and an underwater pipeline desilting system with a suction head is set up in the reservoir area. During the process of pipeline layout, an adverse slope section will appear due to the influence of the dam body. Wang [17] found that in the reverse slope section of a reservoir self-priming pipeline, because the sinking direction of the heavy solid material moves in the opposite direction of flow, it is often the portion part that has a high occurrence of pipe plugging. If the sediment in the reservoir has been silted for a long time, the sediment is a severe concretion [18–21].

In the design of a mud transportation pipeline, to prevent pipeline silting and ensure transportation safety, the design of the pipeline transportation velocity must be greater than the critical non-silting velocity. The critical non-silting velocity, however, is typically determined according to the data of the downslope or the flat slope, and the calculated value is often small when applied to the reverse slope section. From the perspective of an engineering application, the two most important problems in the research of pipeline sediment transport are as follows: (1) the effect of sediment on flow energy consumption and (2) the law of sediment deposition during the process of pipeline transportation. To solve the problem of silting and blocking in pipeline transportation, it is necessary to recognize the law of sediment transport in the reverse pipe section.

This study focuses on the engineering technology demand of mud and clay block transportation in a reverse slope pipe section in a self-priming pipeline desilting project. The present study (1) examines the transport law of sediment (or clay block) using the flow in the pipe section with an adverse slope and (2) determines the relationship between sediment (or clay block) and water flow under different pipe slopes. The results provide a scientific and reasonable design basis for the layout of sediment discharge pipelines in a reservoir desilting project.

## 2. Materials and Methods

### 2.1. Model Design

The experimental device was composed of a muddy water pipe circulation system, a sediment mixing Gaza system, a sand sink, and an iron frame with a slide (Figure 1). The water intake was 50 cm from the ground, the leftmost end was the stirring tank, and the right end of the stirring tank was three sections of steel tubes. The motor (to drive the pump), flow meter, and valve were arranged from left to right along the process.

The rightmost end of the steel pipe was connected with the horizontal section of the plexiglass pipe, and then the horizontal section of the plexiglass pipe was connected with the reverse slope section to the elbow plexiglass pipe. The end of the reverse slope section was then connected with the hose, and the hose was connected to the upper portion of the mixing tank. The length of the plexiglass pipe in the horizontal section was 2 m, and the length of the plexiglass pipe in the reverse slope section was 2.5 m. The horizontal plexiglass pipe and the reverse slope plexiglass pipe were connected using an elbow. A 10 m long plastic hose was used behind the reverse slope plexiglass pipe to make the water return to the mixing tank for circulation and to keep the sediment content in the mixing tank stable.

At the left end of the steel tube was the motor (driven with a pump), at the middle end was the flow meter, and at the right end was a manual valve. The plexiglass pipe in the horizontal section was connected to the steel pipe through a flange, and three pressure sensors were arranged from left to right. The observation section for silting was located 0.5 m from the leftmost end of the horizontal pipe, and the cameras and lighting equipment were arranged below the pipe. The horizontal section and the reverse section of the plexiglass pipe were connected using elbows to realize the slope control of the plexiglass pipe and to minimize leakage (Figure 1, Table 1).

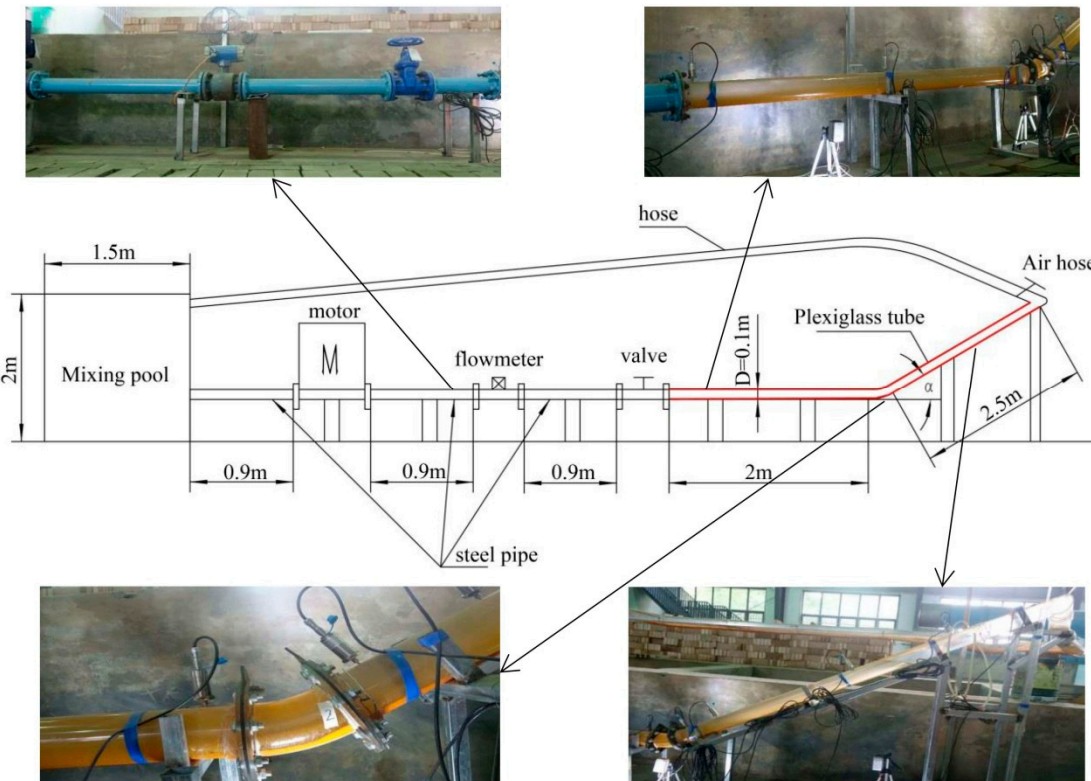

**Figure 1.** The frame diagram of model design.

**Table 1.** Test materials and instruments.

| Items | Materials and Equipment |
|---|---|
| Steel pipe | Inner diameter of 100 mm, and three steel tubes (0.9 m) |
| Hose | Inner diameter of 80 mm and length of 10 m |
| Plexiglass tube | Inner diameter of 100 mm, the length of two glass tubes: 2 m and 2.5 m, and the elbow angle: 30°, 45°, and 60° |
| Flow control | Frequency converter and valve |
| Filming equipment | Video recorder, video camera |
| Flow meter | Electromagnetic flow meter |
| Slope control | Plexiglass pipe elbow |

## 2.2. Experimental Design

The right end of the steel tube was the horizontal section of the glass tube. In the horizontal section of the glass tube, pressure sensor channels 1, 2 and 3 were arranged. Five measuring points on the reverse slope section were also arranged, namely, channels 4, 5, 6, 7 and 8. Because channel 8 was affected by turbulence and negative pressure, it was abandoned. An opening was constructed along the top portion of the reverse slope section where the hose met the channel and the rubber hose was inserted. This was done if the Dong-Hua Test Real Time Data Measurement Analysis Software System (DHDAS) had to be cleared; if not, it was opened and clamped the rest of the time (Figure 2).

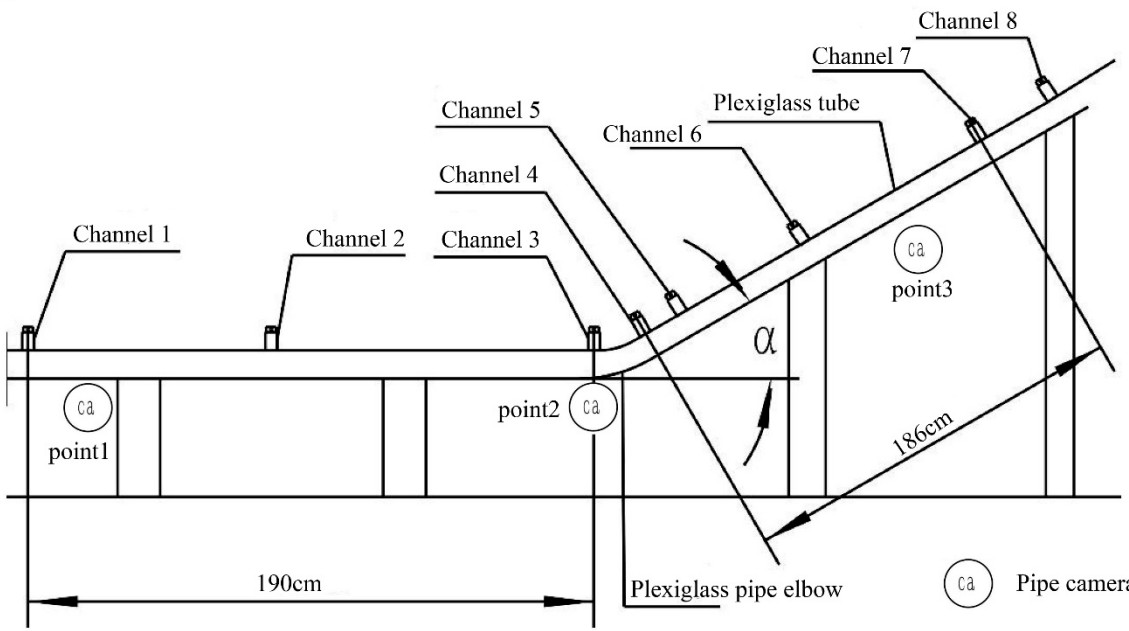

**Figure 2.** Schematic diagram of pipeline pressure measuring point.

### 2.2.1. Test Sand

The sand used in this test was quartz sand with a median particle size of 0.3 mm. The particle size distribution presented a single peak, and the sediment uniformity was relatively high. D03, D06, D10, D16, D25, D50, D75, D84, D90 and D97 were 0.153 mm, 0.170 mm, 0.186 mm, 0.205 mm, 0.229 mm, 0.292 mm, 0.378 mm, 0.427 mm, 0.473 mm and 0.573 mm, respectively (Figure 3).

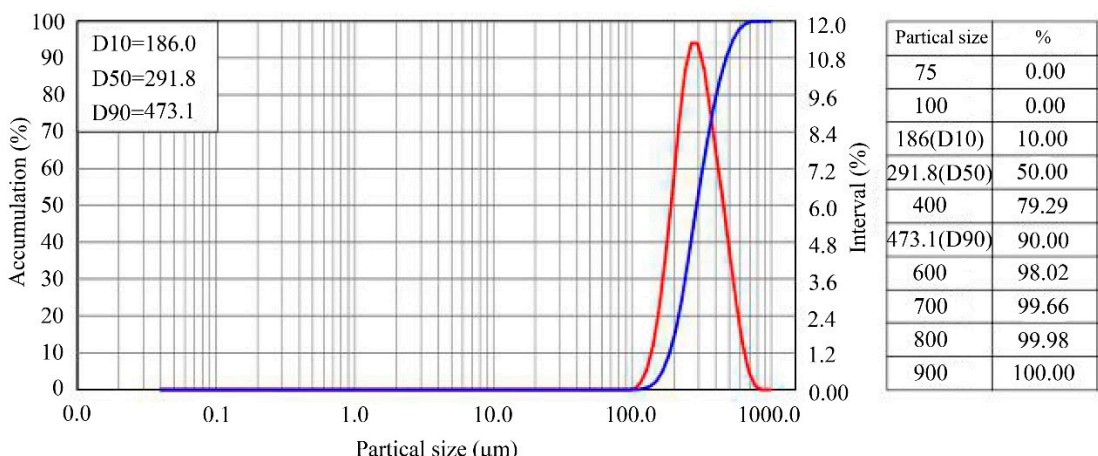

**Figure 3.** Particle size distribution of the test sand (red line, relationship between the sediment particle size and volume; blue line, relationship between the sediment particle size and the cumulative volume).

### 2.2.2. Experimental Arrangement

The reverse slope pipeline (30°, 45°, and 60°) was selected for this study. The sediment concentration ranged from 0 to 270 kg/m³, and the interval of sediment concentration in each group was 30 kg/m³ (Table 2). The flow meter used was an electromagnetic flow meter. The current flow value could be read directly through the electromagnetic flow meter screen after electrification. The range of the flow meter was 0–38 L/s, and the accuracy was 0.01 L/s. A frequency converter was used for flow control. This offered the advantages of energy savings and high precision to fully open

the valve and adopt the frequency converter for flow control. The frequency adjustment range of the frequency converter was 0–50 Hz, and the accuracy was 0.001 Hz.

**Table 2.** Test group arrangements.

| Slope | Sediment Concentration (kg/m$^3$) | Test Groups |
|---|---|---|
| 30°, 45°, 60° | 0 | Three repetitions |
| | 30 | |
| | 60 | |
| | 90 | |
| | 120 | |
| | 150 | |
| | 180 | |
| | 210 | |
| | 240 | |
| | 270 | |

The experimental steps were as follows.

(1) Prior to the experiment, the appropriate amount of sand was added, the blender was started, and the mixture was stirred for half an hour. This ensured that the sediment content in all portions of the mixer reached the same level.

(2) The valve was then opened, and the motor (to drive the pump) was started. This flushed the silt into the glass tube, and then the motor and valve were closed.

(3) Because a siphon formed in a section behind the highest point, the hose was loosened at the upper portion of channel 8. Note, however, that the highest point of the water level had to be higher than the position of channel 7.

(4) The DHDAS system was opened, and the upper hose of channel 8 was closed after the pressure was cleared at each measuring point in the tube.

(5) At the same time, the motor and valve were opened to produce a large flow rate so that the sediment in the measuring section of the pipe would not accumulate. Photos were taken, and the phenomenon and flow rate were recorded. The sediment concentration was measured using a specific gravity bottle, and the DHDAS system was used to measure the pressure at each channel in the pipe.

(6) The motor was adjusted, and the flow rate was reduced to only one test section deposition, two depositions, and three depositions in the tube; step (5) was then repeated.

(7) The motor was turned off, the mixer was turned off, the video and data were backed up, and the tests were transferred to a hard disk.

*2.3. Data Analysis*

When the initial flow was high, no deposition occurred in the tube, and the flow was uniformly and slowly adjusted down. We observed whether or not the sediment in the observation section of the tube remained stationary and recorded the current flow (deposition flow). Critical deposition rate was calculated as follows:

$$U_c = \frac{4Q}{\pi D^2} \tag{1}$$

where $U_c$ is the critical deposition rate (m/s); $Q$ is the sedimentation flow (measured values based on the physical model, m/s$^3$) and $\pi = 3.141$.

In this study, the calculated values of the four classical critical deposition velocity equations were compared with the measured values (Table 3).

**Table 3.** Critical deposition velocity equations.

| Reference | Particle Size (mm) | Pipe Diameter (mm) | Equation | Parameters |
|---|---|---|---|---|
| Shook (1969) [22] | 0.2–5.25 | 80–580 | $U_c = 2.43 \frac{C_v^{1/3}}{C_D^{1/4}} \sqrt{2gD\left(\frac{\rho_s}{\rho_w} - 1\right)}$ | $C_v$—volume concentration |
| Wasp (1977) [23] | 0.25–2.04 | 26.7–139.7 | $U_c = 3.40 C_v^{0.22} \left(\frac{d_s}{D}\right)^{1/6} \sqrt{2gD\left(\frac{\rho_s}{\rho_w} - 1\right)}$ | $C_D$—drag coefficient $g$—9.8 m/s$^2$ $D$—pipe diameter |
| Newitt (1955) [24] | 0.208–3.81 | – | $U_c = 13.88\left(\frac{d_s}{D}\right)^{0.5} C_d^{-0.5} \sqrt{2gD\left(\frac{\rho_s}{\rho_w} - 1\right)}$ | $\rho_w$—liquid density $d_s$—particle size |
| Turian (1987) [25] | 0.02–2.2 | 58–101.6 | $U_c = 1.82 C_v^{0.11}(1 - C_v)^{0.25}\left(\frac{d_s}{D}\right)^{0.06} \sqrt{2gD\left(\frac{\rho_s}{\rho_w} - 1\right)}$ | $\rho_s$—solid density |

The reverse slope pipe head loss was calculated using the Durand and GiBert [26] equations as follows:

$$i_m = i_0 \pm (i_h - i_0)\cos\theta \tag{2}$$

where $i_h$ is the horizontal pipe slurry head loss (kPa/m); $\theta$ is the inverse slope angle (°) and the other symbols have the same meaning as described previously.

Woester and Denny [27] found that the head loss of the solid–liquid two-phase flow in the inclined pipe was equal to the head loss of a vertical pipe and horizontal pipe connected with an inclined pipe as follows:

$$i_{m1} = i_0 + i_h \cos\theta + C_v\left(\frac{\rho_s - \rho_w}{\rho_w}\right)\sin\theta \tag{3}$$

where the symbols have the same meaning as noted earlier.

## 3. Results and Discussion

### 3.1. Influence Factors of the Critical Deposition Rate

When the sediment was deposited, the drag force of the sediment flow was insufficient for the forward movement of the sediment particles. The observation point of the horizontal section was sedimentation observation point one, and the sedimentation law did not change obviously because of the change in the adverse slope dip angle. For deposition observation point two, however, the gravity of the particles remained unchanged as the angle increased, but the vertical component of the lifting force on the water decreased because the flow direction changed. Additionally, because of gravity, the sediment particles slid along the direction of the reverse flow of the pipe and finally slid down to deposition observation point two. Observation point three took more time to produce sedimentation than point one and point two. When the inverse slope angle was 30° and 45°, the sediment was deposited at the bottom of the pipeline at observation point three. When the angle was 60°, if the sediment concentration was low, the pipeline sediment would be suspended sedimentation.

If the sediment concentration was small, the sediment produced pipe suspension deposition within a small area (i.e., the sediment particles had a vortex motion). If the sediment content was large, the sedimentation law of the pipeline was roughly the same as that when the angle was 30° or 45°. When only some sediment was deposited, however, the reverse flow direction dropped compared with that when the angle was 30° or 45° (Figure 4).

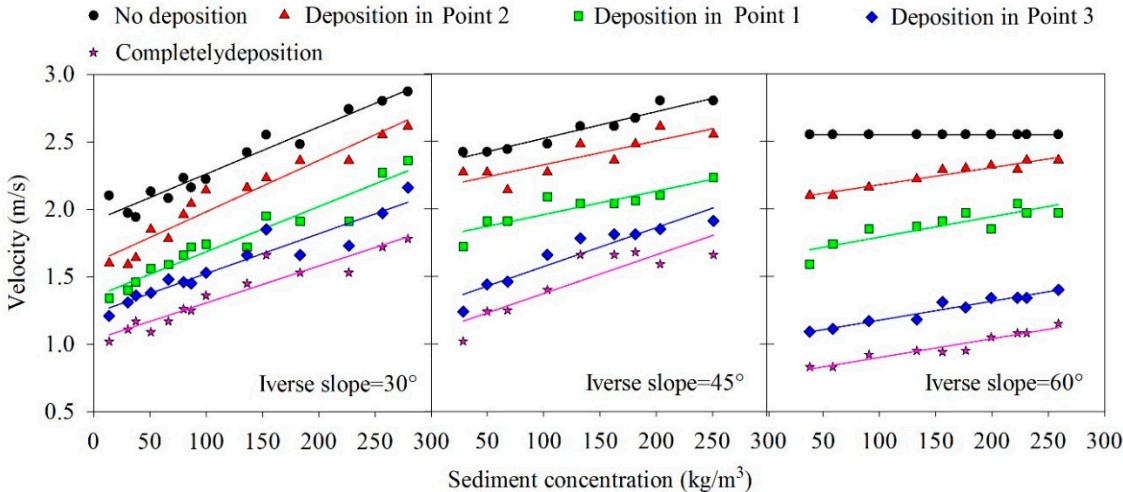

**Figure 4.** Critical deposition velocities under different sediment motion conditions in each group.

The critical deposition velocity at the three observation points increased with an increase in the sediment concentration when the reverse slope remained unchanged. In these three different adverse slopes, the test data curves at the different measuring points were roughly parallel, which indicated that the critical deposition velocity at the different measuring points increased at roughly the same amplitude as the sediment concentration increased, which also verified the conclusion of Xu [28]. Although the increased rate of the critical deposition velocity was roughly the same at the different measuring points in a certain inverse slope pipeline, different adverse slopes had different influences on the rate of increase. This result showed that the rate of increase for a critical deposition velocity of a 60° inverse slope was less than that when the angle was 45° or 30° with an increased rate of sediment concentration, at which point the angle of 30° increased the most.

In this test, quartz sand particles with a median particle size of 0.3 mm were utilized. When the concentration was low and the flow rate was large, most of the sediment particles were suspended, and a few slid. When the concentration was high and the flow was low, the sediment particles were deposited easily. The critical sedimentation velocity of observation point two was higher than that of observation point one and point three. Due to morphological changes in the turning section, the flow strength dissipated rapidly. In the experiment, the pump was shut down, and the sediment on the reverse slope pipe slid along the pipeline quickly. Therefore, the elbow section was prone to siltation, which should be paid enough attention to in practical engineering projects [26].

### 3.2. The Calculation of the Critical Deposition Rate

When the inverse slope of the pipeline was 30°, 45°, or 60°, the critical deposition velocity of the horizontal pipeline measuring point remained close to the Shook equation, and the linear relationship was aligned with the Wasp and Turian equations. When the pipeline inverse slope section was 30°, 45°, or 60°, the linear value of the critical deposition velocity at the measuring point in the reverse slope section was consistent with the Shook equation, the Wasp equation, and the Turian equation, but the actual value was between that calculated using the Wasp [23] and Turian [25] equations (Figure 5).

The Shook equation does not consider the effect of sediment particle size. Considering that the sand particle size used in the test was relatively small, the value particle size was 0.3 mm. If coarse sediment was selected, the measured value of the critical deposition velocity of the test increased, and the measured value curve in the figure moved up, which was closer to the value calculated using the Shook equation. As a result, the value of the Shook equation was not consistent with the actual value, which may have been due to the fact that the Shook's critical deposition velocity equation does not consider the sediment particle size.

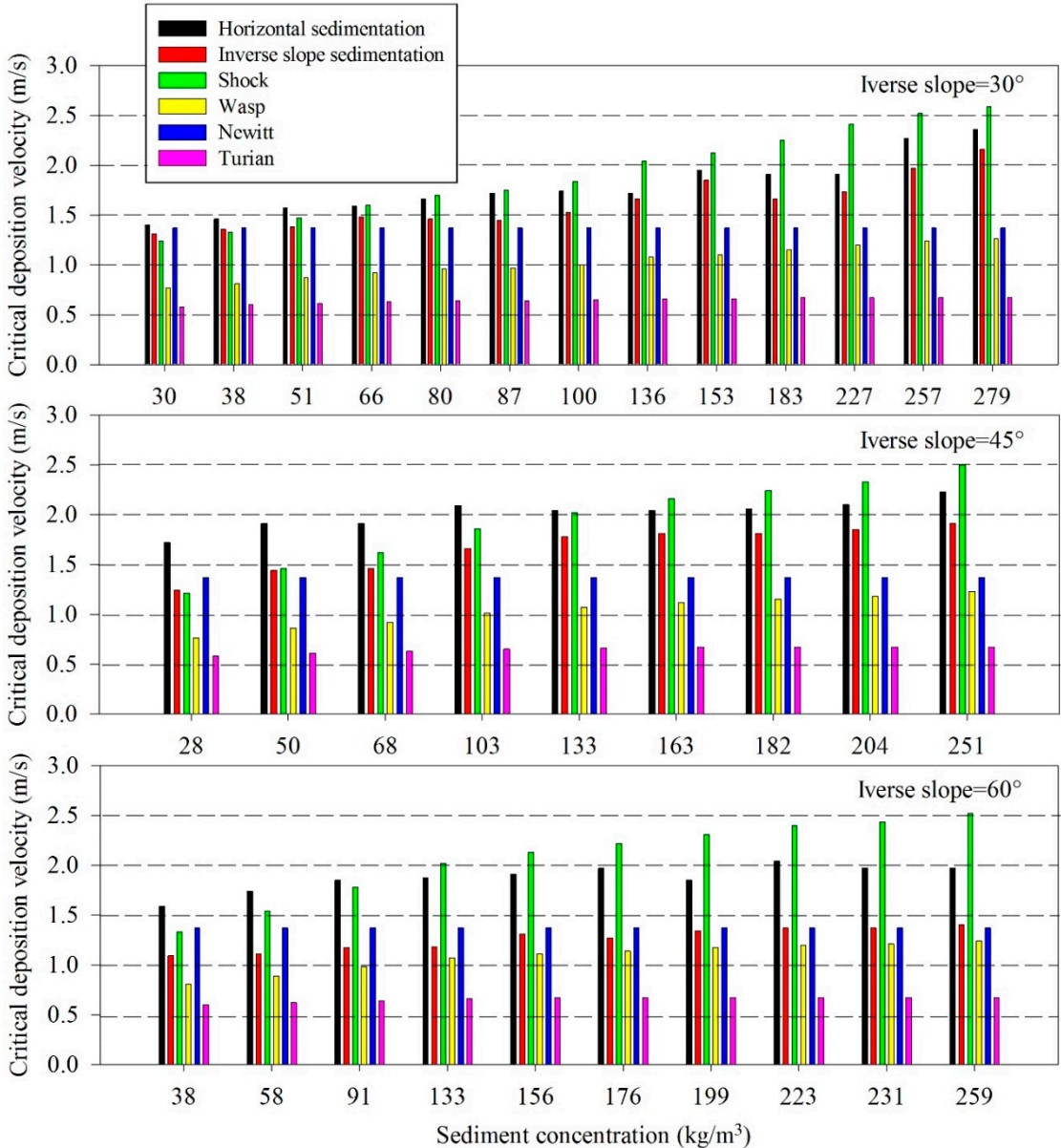

**Figure 5.** The measured value of critical deposition velocity compared with that predicted by different formulas.

The Wasp equation does consider the influence of sediment particle size, sediment density, pipe diameter, and concentration, but the layout of the test pipeline was different from that in this test. In this test, the pipeline layout had a reverse slope dip angle. Previous tests have shown that different test layouts affect the critical deposition velocity of sediment in a pipeline. When there was a reverse slope inclination angle, the critical deposition velocity in the pipeline was larger than that in the horizontal and downslope pipelines. If only horizontal pipelines existed in the pipeline layout in this study, the measured critical deposition velocity in this study would be reduced, which was consistent with the existing research on Wasp.

The Newitt equation considers the influence of sediment particle size, sediment density, and pipe diameter, but does not consider the influence of sediment concentration on the critical deposition rate. The sediment concentration was positively correlated with the critical deposition velocity. The observations from the actual data in this experiment also verified the conclusion of Xu [28]. Because Newitt [24] calculated the value without considering the influence of the sediment concentration,

its critical deposition velocity curve was straight, as shown in Figure 5. Considering this influence, the critical deposition velocity curve would change with a change in the sediment concentration, which was more consistent with the measured value of the critical deposition velocity of the horizontal pipeline in this article [29–31].

Although Turian [25] considers the influence of sediment particle size, sediment density, and pipe diameter, its critical deposition rate does not increase with the volume concentration. Turian [25] believed that when the volume concentration was less than a certain value, the critical deposition velocity increased slightly with an increase in the volume concentration, and when the volume was larger than this fixed value, the critical deposition velocity decreased slightly with an increase in the concentration. In this experiment, the volume concentrations were compared with the sediment concentrations, and the influence of the Turian equation on concentration as the concentration function (dimensionless number) was defined [32]. The change in the concentration function on the sediment concentration is shown in Figure 5. According to existing studies, the critical deposition velocity of the tube sediment increased with an increase in the sediment concentration (or volume concentration), and the Turian studies were inconsistent with these existing studies.

### 3.3. Critical Deposition Velocity of Reverse Slope Section

In the data comparison, although the actual measured data in the test were roughly consistent with the existing results, the specific data were different, so a revised equation was proposed. All of the observations from the Shook equation, the Newitt equation, the Turian equation, and Wasp equations are related to $\sqrt{2gD\left(\frac{\rho_s - \rho_w}{\rho_w}\right)}$. This parameter reflects the influence of the pipe diameter, and the new formula proposed in this study also includes this parameter. In addition, the Wasp equation considers the influence of the sediment particle diameter and the pipeline diameter. The data in this test were in good agreement with the Wasp equation based on Figure 5. Thus, this formula based on the Wasp equation was revised.

To transport certain materials, such as sediment and pulp, density is typically a constant. From the measured data used in this experiment, it was concluded that sediment concentration was positively correlated with critical deposition velocity, whereas the adverse slope was negatively correlated with critical deposition velocity. Combined with the previous formula, after fitting, the critical deposition velocity of the reverse slope pipeline was as follows:

$$U_c = \left(\frac{-0.001\theta^2 + 0.0987\theta - 1.3}{\sin\theta}\right) \times 3.40 C_v \left(\frac{d_s}{D}\right)^{1/6} \sqrt{2gD\left(\frac{\rho_s - \rho_w}{\rho_w}\right)} \tag{4}$$

where the other symbols have the same meaning as provided earlier in this manuscript.

Under adverse slope dip angles of 30°, 45°, and 60°, silting was conducted in accordance with observation point two of the turning section, observation point one of the horizontal section, and observation point three of the reverse slope section. The critical deposition velocities of the pipeline under different sediment concentrations and angles were analyzed. The critical deposition velocity increased with an increase in the sediment concentration. The optimal angle in the reverse slope made the critical deposition velocity of each measured section reach its maximum. When the reverse slope was less than the optimal angle, the critical deposition velocity increased with an increase in the reverse slope. Conversely, when the reverse slope was greater than the optimal angle, the critical deposition velocity decreased. The results obtained by different scholars were summarized and compared with the measured values obtained in this experiment. In this study, the Wasp equation was modified based on the measured value of this experiment. By comparing the calculated value of the new formula with the measured value of this experiment, the results were better than those obtained using the Wasp equation (Figure 6, Table 4).

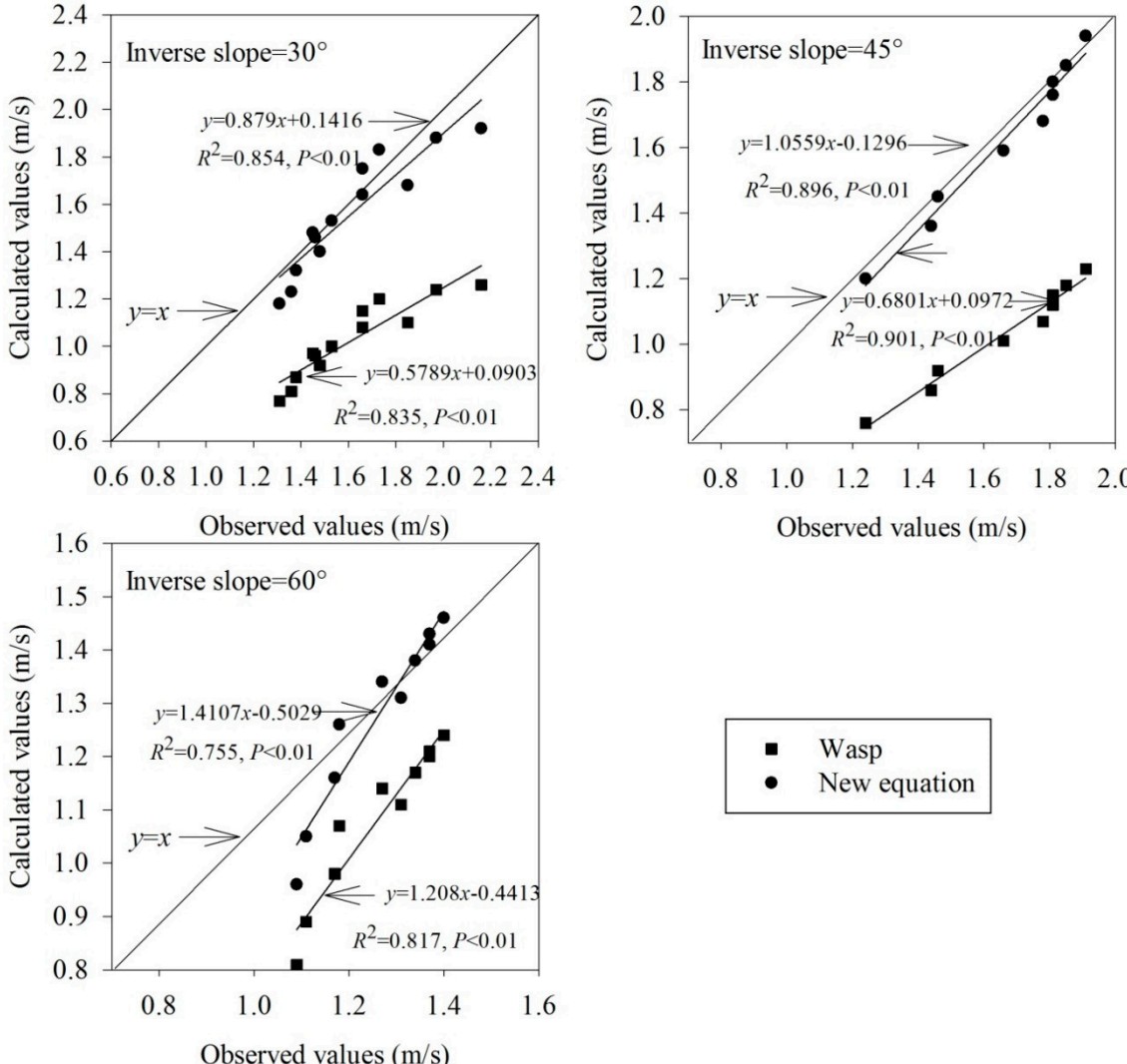

**Figure 6.** The measured critical deposition velocity compared with the new equation and the Wasp equation.

**Table 4.** A comparison of the observed and simulated values of the critical deposition velocities.

| Slope | Observed (m/s) | Simulated (m/s) | | Root Mean Square Error (RMSE) | |
|---|---|---|---|---|---|
| | | New Formula | Wasp | New Formula | Wasp |
| 30° | 1.62 | 1.56 | 1.03 | 3% | 37% |
| 45° | 1.66 | 1.63 | 1.03 | 2% | 38% |
| 60° | 1.26 | 1.28 | 10.8 | 1% | 14% |

*3.4. Influence Factors of the Head Loss in Pipe Sand Transportation*

Under different flow velocities in the 45° and 60° test groups, the hydraulic gradient in the reverse slope section increased with an increase in the sediment concentration. The hydraulic gradient increased with an increase in the flow rate, and the variation law of the hydraulic gradient with different flow rates was consistent. The loss of the solid–liquid two-phase flow head increased with an increase in the velocity (Figure 7). This conclusion was consistent with the research results of Cao [33]. The flow velocity was 0.01 m³/s, 0.015 m³/s, and 0.02 m³/s. It was evident that the flow velocity was important for the solid–liquid two-phase flow head loss. With an increase in the flow velocity, the collision and friction between sediment particles and the friction between sediment particles

and the pipe wall increased, and the energy dissipation also increased. Therefore, the flow velocity increased, and the flow head loss of the solid–liquid two-phase increased.

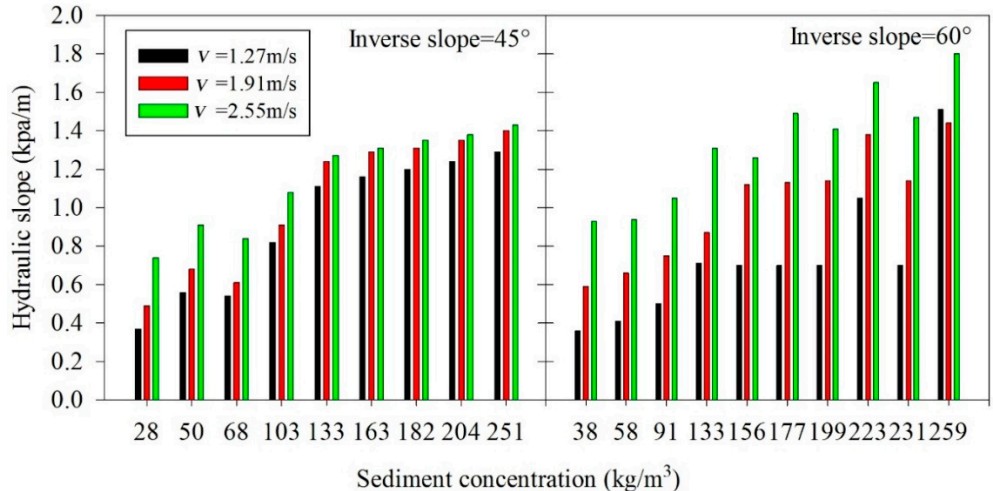

**Figure 7.** The hydraulic gradient of different flow rate varies with sediment concentration.

An optimal angle in the reverse slope made the water head loss of the horizontal pipe and the reverse slope pipe reach a maximum value. When the inverse slope was less than the optimal angle, the head loss increased with an increase in the angle. When the inverse slope was greater than the optimal angle, the head loss decreased with an increase in the angle. This finding also agreed with the research results of Cao [33], Ning [34], and Zhao [35], and in this regard, this experiment played a verification role. Because of the small dip angle, the downward component of sediment gravity along the pipeline increased with an increase in the angle, and a large portion of the water drag force was offset by the gravity component. A large portion of the water flow energy was consumed in the gravitational potential energy loss of solid materials. With an increase in the adverse slope, the gravity component of the vertical pipe slope decreased, and the friction between the solid particles decreased beyond a certain angle, and then the energy consumption was lower than that at this angle [36–38].

The volume concentration of the solid–liquid two-phase flow referred to the volume of the solid phase contained in the solid–liquid two-phase flow per unit volume, which was expressed by the sand content in this experiment. The sand content referred to the weight of the solid phase contained in the unit volume [39–41]. An increase in the sediment concentration resulted in an increase in the number of solid particles per unit volume, an increase in the collision and friction between solid particles, and an increase in the head loss between the solid and liquid two-phase flows. Currently, studies on the specific impact of interparticle collisions and friction on energy dissipation have not been completed, so the energy loss caused by particle collision was ignored.

### 3.5. Calculation of the Reverse Slope Pipe Head Loss

By comparing the actual measured value of this test with the calculated value of the Durand and Woster equations under the reverse slope of 45°, the following was found: (1) the calculated value of Woster equation was much higher than the actual value; (2) when the sediment content was less than 180 kg/m$^3$, the measured value was close to the calculated value of the Durand equation; and (3) when the sediment content was greater than 180 kg/m$^3$, the measured value deviated from the value calculated using the Durand equation (Figure 8).

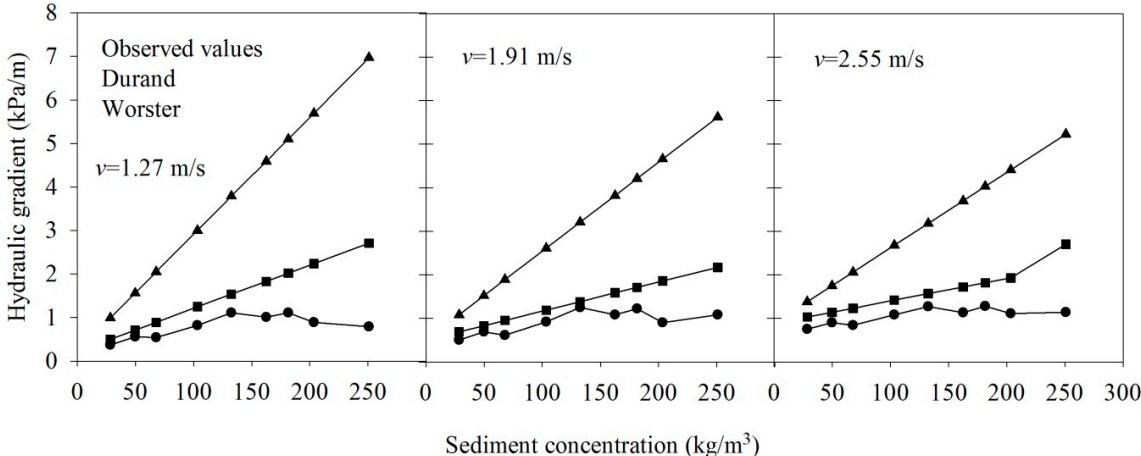

**Figure 8.** Comparison of the observed and simulated values at different flow velocities (inverse slope = 45°).

A comparison between the actual measured value and Durand and Woster equations under the condition of a 60° reverse slope showed that: (1) the calculated value of the Woster equation was greater than the measured value, and (2) the measured values were close to the calculated values of the Durand equation (Figure 9).

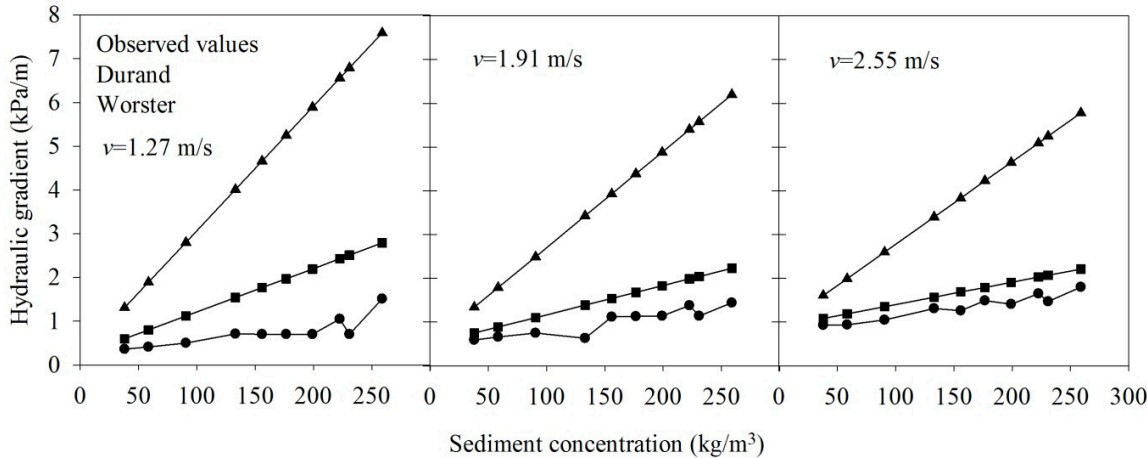

**Figure 9.** Comparison of the observed and simulated values at different flow velocities (inverse slope = 60°).

The measured value and the Woster equation calculated value were far from each other. The Woster equation considered the head loss caused by gravity lift, and the measured value of this test excluded the influence of head loss caused by gravity lift. The measured value was close to the Durand equation, but this result was not completely consistent due to the difference in particle sizes and densities of the solid particles used. In addition, when the sand content at an angle of 45° was greater than 180 kg/m³, the measured value deviated from the calculated value.

## 4. Conclusions

Sediment deposition is an inevitable problem of reservoir engineering on sandy rivers. A large amount of sediment deposition will directly affect the normal operation and service life of a reservoir, and even cause serious events such as reservoir scrapping and dam breaks. According to the physical model, the order of critical deposition velocities at different measuring points was, turning section > horizontal section > reverse slope section. There was an optimal angle between the horizontal section

and the reverse slope section, which made the critical deposition velocity reach the maximum values. The critical deposition velocities of the horizontal section, turning section, and reverse slope section were directly proportional to the sediment concentration. In the current study, the Wasp equation was modified based on the measured data, and the calculated values of the modified equation were in good agreement with the measured values.

This study examined the engineering technical requirements of sediment transport in the reverse slope section of a self-priming pipeline. The law of sediment transport with water flow in a reverse slope pipe section was studied. The interaction mechanism between sediment and water flow under different pipeline slopes was revealed. This study provides a scientific and reasonable design basis for the distribution of sand discharge pipelines for reservoir dredging projects.

**Author Contributions:** Conceptualization, J.W. and S.J.; methodology, L.P.; validation, Z.L.; formal analysis, J.W.; investigation, J.W.; resources, J.W.; data curation, L.P.; writing—original draft preparation, J.W.; writing—review and editing, S.J. and Y.L.; supervision, S.J.; project administration, J.W.; funding acquisition, J.W. All authors contributed to the final version of the manuscript.

**Funding:** This research was funded by the Central Public-Interest Scientific Institution Basic Research Fund, grant number (HKY-JBYW-2018-10), the National Key Research Priorities Program of China, grant number(2018YFC040720203), the Central Public-Interest Scientific Institution Basic Research Fund, grant number (HKY-JBYW-2020-05), Open Project Fund of Key Laboratory of Yellow River Sediment Research, Ministry of Water Resources and the National Natural Science Foundation of China, grant number (31700370).

**Acknowledgments:** We would like to thank the reviewers very much for their valuable comments and suggestions. We also are thankful for the valuable comments of other colleagues and suggestions that have helped improve the manuscript. We thank LetPub (www.letpub.com) for linguistic assistance during the preparation of this manuscript.

**Conflicts of Interest:** The authors declare no conflict of interest.

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
