# Peer review of "Study of the Sediment Transport Law in a Reverse-Slope Section of a Pressurized Pipeline"

_water, doi:10.3390/w12113042_

Round 1

Reviewer 1 Report

no comments

Reviewer 2 Report

This is my second review of this manuscript for the water journal. In the previous version, I recommended major revision and gave 17 comments to the authors. I believe that they did a great deal of work during the amendments and the paper was significantly improved. It should be emphasized that it is also much better read, as the English has been improved. The authors referred to most of the comments I had in the previous version. However, they have omitted the following two, which I would like to be corrected before the work is published.

Lines 14-19: The abstract begins with a run-on sentence, which should be omitted.

Lines 118-119. Although they calculated D50, there still is lack of sorting in the paper.

Author Response

This manuscript is a resubmission of an earlier submission. The following is a list of the peer review reports and author responses from that submission.

Round 1

Reviewer 1 Report

I have several comments before possible publication of this manuscript.

  1. Lines 14-19: The abstract begins with a run-on sentence containing several other grammatical errors. Moreover, the entire manuscript follows suit. I strongly recommend that the authors seek out extremely thorough editorial support on any revised draft from a native English speaker.
  2. Lines 19 and 21. There are repetitions that should not be the case in scientific language
  3. Lines 38-40. This sentence adds nothing to this work and should be deleted.
  4. Lines 45-48. This sentence is too long and should be changed, and the content should be communicated more concisely and simply.
  5. Line 58. I am not sure if the word hardened is used correctly here.
  6. Line 58. This sentence is unnecessary.
  7. Line 63. the statement “remains scare” is odd and seems inaccurate. You just listed several studies related to the subject, and listed several formulae associated with this topic.
  8. Lines 73 and 75. There are repetitions that should not be the case in scientific language
  9. Line 80. Why “about”?
  10. Lines 86 and 88. There are repetitions “we connected” that should not be the case in scientific language
  11. Lines 117/118. Please calculated for example: sorting of your material and add info to the paper.
  12. Lines 123 and 125. The same info is presented in two sentences. It seems the authors sent the article in a hurry ... Make sure each author is fully involved in the cycle of editing. I find it useful for each author to act as a ‘hostile’ reviewer.
  13. Lines 174, 186, 325, 436… There is no statistical analysis in the paper, but the authors claimed that they observed significant differences between obtained data. At least ANOVA is required for proof of your findings.
  14. Line 198. There is lack of word “median” in this sentence.
  15. Section 3.3 is very difficult to follow. There are editorial mistakes. It should be removed from the paper or deeply improved. Moreover, the authors claimed that “test were in good agreement” and that data were correlated. I would like to see those results.
  16. Lines 290 and 370. Where are the results of error value presented?
  17. Conclusion section. After reading of the whole manuscript I do not know what I learnt from this paper. Please clearly develop this section by improving what new information was added to our knowledge.

Reviewer 2 Report

The idea of the paer and the experiment looks fine. However plese explain what for you did this experiment, for whom and why? Is it better using your formula or that those forlulae already existed? What is a practical advantage of using your formula that those already existed? Who would benefit using your formula and what way somebody (or a company) benefit? What is really new in your formula and how industry would benefit of it? Solid and convincing arguments are expected.

Reviewer 3 Report

Dear authors,

the investigated topic is interesting and I appreciate the clear practical background of your experiments.

However, the way you presented it is below an acceptable level. This goes through the whole paper.

Already at the introduction an interested reader will not understand  how a "reverse slope section of reservoir self-priming pipeline" should look like. You need to explain the idea and technical implementation with a descriptive scheme.

Following you discuss the problem of mud blocks with a diameter of 3-5 cm, which are easily blocked. However in the experimental design you do not investigate these but silt/sand with particle size between 75 to 1000 µm. So, the causal relationship is not given.

Me and my team tried hard to understand your experimental design, without complete success. The same appllies for the data analysis (see remarks and question in the pdf of the paper).

The discussion is really exhaustive, describing qualitatively what can be seen in the diagrams. A discussion should try understand/enlighten the processes behind.

Table 4 does not belong to the paper at all. This is a clear hint, how careless you have written this paper.

Provided, the experimental design and data analysis are scientifically correct (which I cannot proove, based on the rather weird description),  then the proposed new sedimentation formula could be really a reasonable contribution for the community. However, I haven't understand how you derived the formula. Is this simply an emperic mathematical approximation to the measured data, or are there any phyiscal laws behind.

The validation of the equation with new data is generally the right approach. However a simple visual judgement is not sufficient. You have to apply objective functions, like RMSE and give the R² for the regression lines. The regression lines are far from the y=x line. However, these deviation require a thorough discussion, which I missed.

and so on....

We gave you a lot of comments. questions and hints for improvement directly in the text.

I will reject the paper in its current form. If you completely revise it, with strong causal chain (background, objective, appropriate experimental design, discussion), your investigation might be meaningful and worth for a new submission. Your English requires significant improvement, more in the diction than in grammar. Many sentences, I read several times to get the idea.
